# BSO: Binary Spiking Online Optimization Algorithm

Yu Liang [1]   Yu Yang [1]   Wenjie Wei [1]   Ammar Belatreche [2]   Shuai Wang [1]   Malu Zhang [1]   Yang Yang [1]

## Abstract

Binary Spiking Neural Networks (BSNNs) offer promising efficiency advantages for resource-constrained computing. However, their training algorithms often require substantial memory overhead due to latent weights storage and temporal processing requirements. To address this issue, we propose Binary Spiking Online (BSO) optimization algorithm, a novel online training algorithm that significantly reduces training memory. BSO directly updates weights through flip signals under the online training framework. These signals are triggered when the product of gradient momentum and weights exceeds a threshold, eliminating the need for latent weights during training. To enhance performance, we propose T-BSO, a temporal-aware variant that leverages the inherent temporal dynamics of BSNNs by capturing gradient information across time steps for adaptive threshold adjustment. Theoretical analysis establishes convergence guarantees for both BSO and T-BSO, with formal regret bounds characterizing their convergence rates. Extensive experiments demonstrate that both BSO and T-BSO achieve superior optimization performance compared to existing training methods for BSNNs. The codes are available at https://github.com/hamingsi/BSO.

## 1. Introduction

Spiking Neural Networks (SNNs), as the third generation of neural network paradigms (Maass, 1997; Roy et al., 2019b), have garnered significant attention in machine learning attributed to their bio-inspired mechanisms and energy efficiency advantages. Building upon this foundation, Binary SNNs (BSNNs) further enhance computational efficiency by incorporating binary weights representations, offering

a promising solution for resource-constrained edge computing scenarios (Srinivasan & Roy, 2019; Lu & Sengupta, 2020; Kheradpisheh et al., 2022). Despite these notable advantages, BSNNs predominantly utilize training algorithms inherited from full-precision SNNs.

Current BSNN training methods fall into two main categories: ANN-SNN conversion and direct training. ANN-SNN conversion obtains BSNN weights from pre-trained Artificial Neural Networks (ANNs) (Rueckauer et al., 2017; Roy et al., 2019a; Wang et al., 2020; Yoo & Jeong, 2023). However, these approaches typically require long simulation time steps to match ANN performance, inevitably increasing energy consumption. Moreover, it cannot handle sequential data. To address these limitations, researchers have developed direct training algorithms for BSNNs, with surrogate gradient-based backpropagation through time (BPTT) emerging as a predominant training paradigm (Deng et al., 2021; Pei et al., 2023; Wei et al., 2025). Unfortunately, this methodology requires maintaining both computational graphs and latent weights for BSNNs, resulting in significant memory and computation overhead. Consequently, this resource-intensive training process inherently contradicts the efficiency advantages of BSNNs. This motivates us to raise an intriguing question: *Can we leverage the substantial efficiency advantages of BSNNs not only during the forward pass but also throughout the learning process?*

Recently, online training algorithms for SNNs have gained prominence for their temporal independence, which substantially reduces training costs in memory (Xiao et al., 2022; Meng et al., 2023; Jiang et al.). However, existing SNN online training algorithms are predominantly designed for general situations. Given the efficiency advantages of BSNNs in resource-constrained environments, the integration of online training presents a promising direction in efficient neuromorphic computing. However, this integration faces a fundamental challenge: the storage overhead of latent weights in existing BSNN training methods conflicts with online training's memory efficiency objectives. This contradiction underscores the urgent need to develop specialized online training techniques tailored for BSNNs.

To address these challenges, we propose a Binary Spiking Online (BSO) optimization algorithm that leverages BSNN's efficiency advantages in both forward and back-

---

[1]University of Electronic Science and Technology of China [2]Northumbria University. Correspondence to: Malu Zhang <maluzhang@uestc.edu.cn>.

*Proceedings of the 42nd International Conference on Machine Learning*, Vancouver, Canada. PMLR 267, 2025. Copyright 2025 by the author(s).

ward propagation. BSO offers two key advantages: first, its memory requirements are orthogonal to time steps, significantly reducing memory and computational overhead; second, it updates weights through sign flips, eliminating the need for latent weights. Furthermore, to enhance performance by leveraging the inherent temporal dynamics of SNNs, we introduce a temporal-aware variant called T-BSO, which effectively integrates gradient information across each time step.

**Contributions.** In summary, our key contributions are summarized as follows:

- We introduce BSO, the first online training optimization algorithm tailored for BSNNs. BSO significantly reduces memory overhead through two advantages: (a) ensuring memory requirements orthogonal to time steps and (b) eliminating latent weights storage via gradient-dominated flipping.

- We propose T-BSO, an enhanced version of BSO that significantly improves performance by effectively exploiting SNNs' temporal dynamics, capturing the second-order gradient moment across time steps for adaptive threshold optimization.

- We rigorously prove the convergence of both BSO and T-BSO algorithms by deriving their theoretical regret bounds. Our comprehensive analysis demonstrates that the parameter updates in BSO and T-BSO maintain substantial directional consistency with convergence.

- Our BSO and T-BSO demonstrate superior performance on both large-scale static and neuromorphic datasets across CIFAR-10, CIFAR-100, ImageNet, and CIFAR10-DVS. Experimental results verify time-independent training memory requirements, substantially reducing training overhead.

## 2. Related Work

### 2.1. Learning algorithm for BSNNs

Training algorithms for BSNNs fall into two main categories: ANN-SNN conversion methods and direct training methods. The ANN-SNN conversion method trains an ANN first, then directly transfers the weights to a BSNN. (Roy et al., 2019a) first analyze the combination of different binary neurons and binarized weight methods to train an ANN, and then obtain a deep SNN with binary stochastic activations through a conversion approach. Similarly, (Lu & Sengupta, 2020) utilize standard training techniques to train a binary convolutional model, then generate a binary SNN via a conversion process. Furthermore, (Wang et al., 2020) examine the relationship between weights and spiking neuron

thresholds, proposing a weight-threshold balancing conversion method to reduce errors during conversion. However, these converted SNNs suffer from performance degradation and long latency. To address these limitations, researchers have developed direct training methods for BSNNs. (Qiao et al., 2021) employ a surrogate gradient method to train a hardware-friendly BSNN for processing temporal neuromorphic data. (Kheradpisheh et al., 2022) introduce BS4NN, a temporal-encoded BSNN where each neuron fires at most one spike, providing an energy-efficient event-driven learning approach. To further reduce the quantization error and improve performance, some studies have introduced different strategies to enhance performance, such as alternating direction methods (Deng et al., 2021), accuracy loss estimator (Pei et al., 2023), improved activation function (Hu et al., 2024), weight-spike regulation (Wei et al., 2024; Wang et al., 2025), and adaptive gradient modulation (Liang et al., 2025). Recently, researchers have also designed quantization strategies for spiking transformers, achieving competitive results on vision tasks (Qiu et al., 2025; Cao et al., 2025).

As described above, existing learning algorithms for BSNNs are adapted from full-precision SNNs. While these methods leverage BSNN's energy efficiency in forward propagation, they fail to exploit its binary weight characteristics during the training process. This motivates us to design a dedicated training algorithm for BSNN that leverages the weight sign flipping property for parameter updates and eliminates latent weight reliance, enabling energy efficiency in both forward and backward propagation.

### 2.2. Online Training Method in SNNs

In recent years, significant progress has been made in online training algorithms for SNNs to achieve memory-efficient and online training (Zenke & Ganguli, 2018; Bellec et al., 2018; Bohnstingl et al., 2022). Kaiser et al. (2020) proposes local training methods that disregard temporal dependencies while Yin et al. (2023) adapts the approach from (Kag & Saligrama, 2021) using gated neuron models. Notably, OTTT (Xiao et al., 2022) extends the applicability of online training methods to address large-scale computational tasks. SLTT (Meng et al., 2023) establishes that temporal domain gradients contribute minimally to SNN training and proposes their elimination for improved efficiency. NDOT (Jiang et al.) introduces novel gradient estimation techniques based on neuronal dynamic algorithm. These approaches collectively offer diverse solutions for reducing memory overhead during SNN training. However, existing online training methods predominantly focus on full-precision SNNs, leaving a notable research gap in effectively integrating online training with BSNNs. This limitation significantly constrains the training and deployment of BSNNs in resource-constrained scenarios, particularly in practical applications demanding a low memory footprint.

# 3. Preliminaries

## 3.1. Binary Spiking Neural Networks

Spiking neurons form the basis of neural information transmission through spike trains. In BSNNs, these neurons replicate the behavior of a biological neuron which integrates input spikes into its membrane potential $u$ and fires a spike only when $u$ exceeds a threshold. We consider the Leaky Integrate-and-Fire (LIF) model, which describes the dynamics of the membrane potentials as follows:

$$\tau \frac{du(t)}{dt} = -(u(t) - u_{rest}) + R \cdot I(t), \quad u(t) < V_{th}, \quad (1)$$

where $I(t)$ is the input current, $V_{th}$ is the threshold and $\tau$ and $R$ are resistance and time constant respectively. A spike is generated when $u(t)$ reaches $V_{th}$ at time $t^f$, and $u(t)$ is reset to the resting potential $u_{rest}$, which is often zero. The spike train is defined using the Dirac delta function: $s(t) = \sum_t \delta(t - t^f)$.

Spiking neural networks comprise multiple layers of interconnected neurons with associated weights. The input current at layer $l$ and time $t$ follows $I^l[t] = \sum W^l s^{l-1}[t]$, where $W^l$ represents connection weight and $s^{l-1}[t]$ denotes spike outputs from the previous layer. Typically, the discrete method is employed to discretize the dynamic equations of LIF models, resulting in the following iterative form:

$$u^l[t] = \lambda(u^l[t-1] - V_{th}s^l[t-1]) + W^l s^{l-1}[t], \quad (2)$$

$$s^l[t] = H(u^l[t] - V_{th}), \quad (3)$$

where $H(\cdot)$ is the Heaviside step function, $s^l[t]$ and $u^l[t]$ are the spike train and membrane potential at discrete time-step $t$ for layer $l$, and $\lambda$ is a leaky term (typically taken as $1 - \frac{1}{\tau}$). The reset operation is implemented by subtracting the threshold $V_{th}$.

To achieve significant model compression, previous BSNNs methods adopt the sign function during the forward process to convert $W^l$ in Eq.(2) into binary representations. Therefore, the binarization on $w \in W^l$ can be formulated:

$$w_b = \text{sign}(w) = \begin{cases} -1, & \text{if } \omega < 0, \\ +1, & \text{otherwise}, \end{cases} \quad (4)$$

where $w_b$ is a binary weight. Note that the real-valued weights $w$ are maintained as latent weights to accumulate pseudo-gradients. In the backward pass, the straight-through estimator (STE) is employed to approximate the gradient of the sign function (Bengio et al., 2013), i.e., $\frac{\partial sign(\cdot)}{\partial W^l} = 1_{|W^l| \leq 1}$. However, maintaining latent weights incurs additional memory overhead and complicates the optimization process of BSNNs, which will be discussed in Section 4.1.

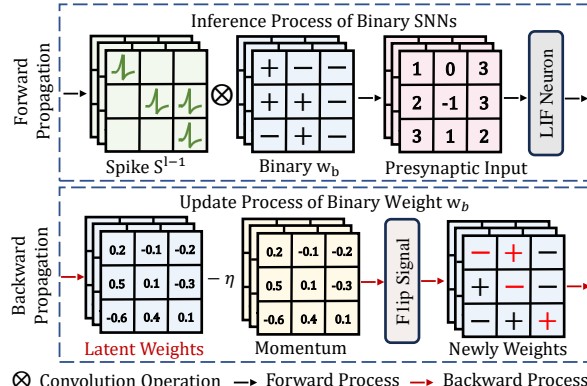

*Figure 1.* Forward and backward propagation of BSNNs under BPTT method.

## 3.2. Online Training Framework

BPTT method unfolds the iterative update equation in Eq.(2) and backpropagates along the computational graph. The gradients with $T$ time steps are calculated by:

$$\frac{\partial \mathcal{L}}{\partial W^l} = \sum_{t=1}^{T} \zeta^l[t] \left( \frac{\partial u^l[t]}{\partial W_l} + \sum_{k<t} \prod_{i=k}^{t-1} \right.$$
$$\left. \left( \frac{\partial u^l[i+1]}{\partial s^l[i]} \frac{\partial s^l[i]}{\partial u^l[i]} + \frac{\partial u^l[i+1]}{\partial u^l[i]} \right) \frac{\partial u^l[k]}{\partial W^l} \right), \quad (5)$$

where $\mathcal{L}$ is the loss, $W^l$ is the connection weight from layer $l-1$ to $l$ and $\zeta^l[t] \triangleq \frac{\partial \mathcal{L}}{\partial s^l[t]} \frac{\partial s^l[t]}{\partial u^l[t]}$ is the gradient for $u^l[t]$. The non-differentiable terms $\frac{\partial s^l[t]}{\partial u^l[t]}$ will be replaced by surrogate functions, i.e. derivatives of rectangular or sigmoid functions (Wu et al., 2018).

For online training framework, we decouple the full gradients into temporal components and spatial components to enable the online gradient calculation. All temporal dependencies are primarily manifested in the spiking neuron dynamics, i.e. $\frac{\partial u^l[i+1]}{\partial s^l[i]} \frac{\partial s^l[i]}{\partial u^l[i]}$ and $\frac{\partial u^l[i+1]}{\partial u^l[i]}$ in Eq.(5). We define the instantaneous gradient at time $t$ as:

$$\nabla_{W^l} L[t] = \zeta^l[t] \hat{a}^l[t]^T, \quad (6)$$

where $\hat{a}^l[t]$ denotes the presynaptic activities that can be tracked recursively during forward propagation:

$$\hat{a}^l[t] = \mu^l[t] \hat{a}[t-1] + s^l[t]. \quad (7)$$

When $\mu^l[t]$ is the constant $\lambda$, the OTTT (Xiao et al., 2022) is obtained, while $\mu^l[t] = \frac{u^l[t] - V_{th}s^l[t]}{u^l[t-1] - V_{th}s^l[t-1]}$ leads to the derivation of the NDOT (Jiang et al.). The total loss $\mathcal{L}$ is defined as the sum of instantaneous losses over time steps:

$$L = \sum_{t=1}^{T} L[t] = \sum_{t=1}^{T} \mathcal{L}(s^N[t], y), \quad (8)$$

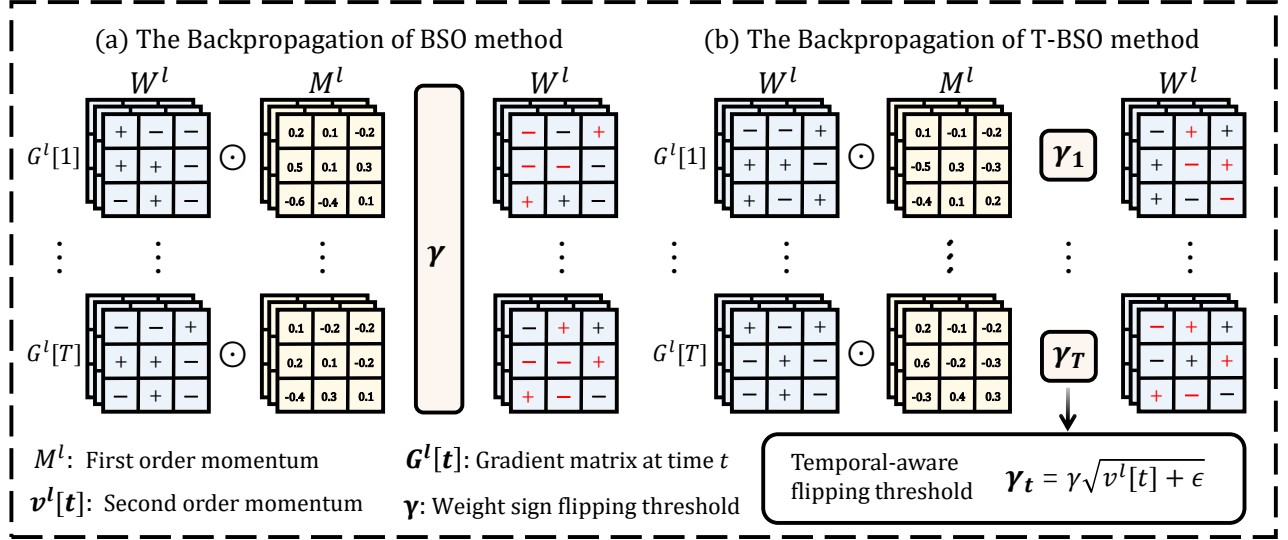

*Figure 2.* Weight update strategies of BSO and T-BSO during backpropagation. (a) BSO employs the same flipping threshold $\gamma$ at each time step, updating synapse weights through the dot product of momentum $M^l$ and Binary $W^l$ in BSNNs. (b) T-BSO incorporates second-order momentum $v^l[t]$ into $\gamma$, thereby achieving temporal dynamic allocation with Temporal-aware flipping threshold $\gamma_t$.

$\mathcal{L}(\cdot)$ can be the cross-entropy loss, and $y$ is the label. Under this framework, parameters can be updated either immediately at each time step (real-time update) or after accumulating gradients over all time steps (accumulated update):

$$W^l \leftarrow W^l - \eta \nabla_{W^l} L[t] \text{ (real-time)}, \qquad (9)$$

$$W^l \leftarrow W^l - \eta \sum_{t=1}^{T} \nabla_{W^l} L[t] \text{ (accumulated)}, \qquad (10)$$

where $\eta$ is the learning rate. This framework enables forward-in-time learning with only constant memory costs, avoiding the significant memory consumption.

## 4. Method

In this section, we first analyze the behavior of latent weights in BSNNs and introduce the BSO optimization algorithm. Secondly, drawing inspiration from the temporal dynamics of neurons, we propose the Temporal-aware BSO to capture temporal gradient information. Finally, we provide an analysis of convergence and computational complexity.

### 4.1. Observation from Latent Weights

To ensure compatibility with existing backpropagation frameworks (e.g., BPTT, OTTT), BSNNs employ latent weights and momentum (Sutskever et al., 2013) to guide binary weight sign updates during training. However, due to the binary paradigm of BSNNs weights, the actual update process only determines which weight signs require flipping, independent of the specific magnitudes of latent weights. As shown in Figure 1, the latent weights serve only

as indicators of sign flipping, and their precise values are not that important for the update mechanism.

This observation reveals a fundamental critical insight: network behavior modifications in BSNNs are fundamentally governed by the frequency and timing of weight sign flips rather than by the continuous optimization of latent weight magnitudes. Consequently, we can explore innovative weight update strategies that directly generate flip signals. This approach bypasses latent weight representations entirely and thereby significantly reduces BSNN training memory overhead. One intuitive approach involves comparing gradient magnitudes against fixed thresholds to generate flip signals. Therefore, this motivates us to explore more stable and efficient optimization strategies that better align with the inherent spatiotemporal characteristics of BSNNs.

### 4.2. Binary Spiking Online Optimization

As discussed in Section 4.1, we aim to completely eliminate latent weights to further significantly reduce the training memory overhead of BSNNs. To this end, we propose a novel momentum-based gradient accumulation strategy combined with threshold-controlled weight updates: BSO algorithm. It leverages momentum to accumulate instantaneous gradients and generate stable weight sign flip signals. At each time step $t$, BSO updates the momentum $M^l$ for layer $l$ based on the computed instantaneous gradient:

$$M^l \leftarrow \beta \cdot M^l + (1 - \beta) \cdot G^l[t], \qquad (11)$$

where $G^l[t] = \nabla_{W^l} L[t]$ and the hyper-parameter $\beta$ control the exponential decay rates of the momentum. When the

value of $\beta$ is relatively large, a greater weight is assigned to historical gradients, making the momentum less sensitive to the gradients in the current time step. The weight update mechanism is guided by the following equation:

$$W^l \leftarrow -\text{sign}(W^l \odot M^l - \gamma) \odot W^l, \quad (12)$$

As shown in Fig.2, a flip signal is triggered only when the element-wise product of $W^l$ and $M^l$ exceeds the threshold $\gamma$, ensuring alignment between the weight and gradient momentum directions. This selective flipping mechanism enhances optimization stability by flipping weights only when their direction aligns with the gradient momentum. Conversely, when their signs differ, the weight sign is preserved to prevent updates along gradient ascent directions, thereby avoiding potential optimization divergence.

Compared to BPTT, BSO eliminates the need for storing extensive computational histories and completely removes dependency on latent variables. This dual optimization approach enables BSO to achieve both computational efficiency and significantly reduced resource consumption during training. However, BSO's uniform gradient treatment across time steps overlooks valuable temporal information, leading to the loss of BSNN's inherent temporal dynamics. Therefore, this motivates us to further explore advanced extensions of BSO for temporal dynamics.

### 4.3. Temporal-aware BSO

As described above, BSNNs exhibit temporal dynamics where neuronal states create dependencies between past and current inputs. Each time step generates unique components from presynaptic activity $\hat{a}^l[t]$, causing temporal variation in gradient distributions. Our analysis in Figure 3 confirms this variability is driven by spiking activity and historical states, highlighting the importance of considering individual time step contributions for flip signal generation. While $M^l$ captures gradient information during training, extending it to a temporal-aware representation $M^l[t]$ would better account for SNN dynamics. However, this approach incurs memory costs that scale linearly with time steps. Inspired by adaptive algorithms, we propose a temporal variant to address this challenge: Temporal-aware BSO. The algorithm update the

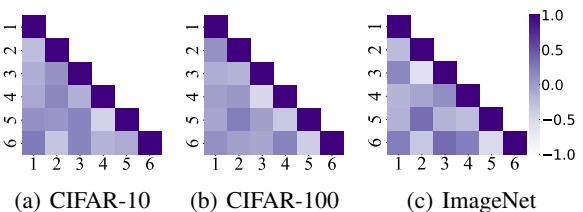

(a) CIFAR-10    (b) CIFAR-100    (c) ImageNet

*Figure 3.* The cosine similarity between the gradients across time step. Both horizontal and vertical axes represent time steps.

first and second order moment of the gradient as follows:

$$\begin{aligned} M^l &\leftarrow \beta_1 \cdot M^l + (1-\beta_1) \cdot G[t], \\ v^l[t] &\leftarrow \beta_2 \cdot v^l[t] + (1-\beta_2) \cdot \text{mean}(G^2[t]), \end{aligned} \quad (13)$$

where the mean function is operated at each layer $l$ and hyper-parameters $\beta_1, \beta_2$ are the decay rates for momentum. We define the temporal-aware threshold as $\gamma\sqrt{v^l[t]+\epsilon}$, where $\epsilon$ is a small constant to prevent zero threshold, thereby enhancing training stability. It accounts for the inherent temporal characteristics of BSNNs by considering gradients across different time steps. This variant avoids the linear growth of training memory cost with time steps by averaging gradients across different time steps. The weight update rule will be transferd as:

$$W^l \leftarrow -\text{sign}(W^l \odot M^l - \gamma\sqrt{v^l[t]+\epsilon}) \odot W^l. \quad (14)$$

Second-order moments $v^l[t]$ are crucial for capturing gradient magnitude variations across temporal dynamics in BSNNs. While first-order moments provide directional guidance, second-order moments adapt to the scale and variance of gradients at each time step. This temporal adaptation enables more precise threshold estimation through $\sqrt{v^l[t]+\epsilon}$, which automatically decreases in regions with consistently small gradients to facilitate weight flipping, while increasing in high-gradient areas to prevent oscillatory updates. This mechanism effectively responds to the heterogeneous gradient distributions that naturally occur across different time steps in spiking neural networks.

Compared to BSO, T-BSO algorithm significantly enhances performance by incorporating temporal gradient dynamics to intelligently control weight sign flips. A key advantage of T-BSO is its highly efficient memory utilization—despite tracking temporal information, it effectively maintains computational efficiency by averaging second-order moments across time steps. The comprehensive details of our approach are provided in Algorithm 1.

### 4.4. Convergence and Complexity Analysis

**Convergence analysis** We analyze the convergence of BSO and T-BSO using the online learning framework proposed in (Zinkevich, 2003). Consider an online optimization scenario where at each round $k \in 1, 2, ..., \mathcal{T}$, an algorithm predicts the parameter $w_k$ and incurs a loss based on a convex cost function $f_k(w)$. The sequence $\{f_k\}_{k=1}^{\mathcal{T}}$ is arbitrary and unknown in advance.

**Theorem 4.1.** *Assume that the function $f_k$ has bounded gradients, $\|\nabla f_k(w)\|_2 \leq G, \|\nabla f_k(w)\|_\infty \leq G_\infty$ for all $w \in \mathbb{R}^d$. Let $\gamma$ and $\beta$ decay by $\sqrt{k}$. BSO achieves the following guarantee for all $k \geq 1$.*

$$R_{\mathcal{T}} \leq 2\sum_{k=1}^{\mathcal{T}} \gamma_k + 2\sum_{k=1}^{T} |M_k|_\infty = O(\sqrt{\mathcal{T}}). \quad (15)$$

*Table 1.* Memory complexity of BSNNs training methods.

| METHODS | WEIGHT STORAGE | COMPUTATION STATE |
|---------|----------------|-------------------|
| BPTT | $O(n^2L)$ | $O(n^2L + nLT)$ |
| OTTT | $O(n^2L)$ | $O(n^2L + nL)$ |
| BSO | $O(n^2L/32)$ | $O(n^2L + nL)$ |
| T-BSO | | $O(n^2L + LT + nL)$ |

*Let $\beta_1, \beta_2$ and $\gamma$ decay by $\sqrt{k}$, the T-BSO achieves the following guarantee for all $k \geq 1$.*

$$R_{\mathcal{T}} \leq 2\sum_{k=1}^{\mathcal{T}} \gamma_k + 2\sum_{k=1}^{\mathcal{T}} |M_k/\sqrt{v_k + \epsilon}|_\infty = O(\mathcal{T}^{3/4}). \tag{16}$$

Our Theorem 4.1 demonstrates that when the gradients of BSNNs are bounded, the summation terms in BSO and T-BSO are respectively bounded by $\sqrt{\mathcal{T}}$ and $\mathcal{T}^{3/4}$. The gradual decay of $\beta_1$, $\beta_2$, and $\gamma$ is essential for our theoretical analysis and is consistent with previous empirical findings. For example, Sutskever et al. (2013) suggests that decreasing the momentum coefficient towards the end of training can improve convergence.

**Corollary 4.2.** *Assume that the function $f_k$ has bounded gradients, $\|\nabla f_k(w)\|_2 \leq G, \|\nabla f_k(w)\|_\infty \leq G_\infty$ for all $w \in \mathbb{R}^d$. BSO achieves the following gurantee, for all $\mathcal{T} \geq 1$.*

$$\frac{R(\mathcal{T})}{\mathcal{T}} = O(\frac{1}{\sqrt{\mathcal{T}}}), \tag{17}$$

*while T-BSO achieves the following gurantee, for all $\mathcal{T} \geq 1$.*

$$\frac{R(\mathcal{T})}{\mathcal{T}} = O(\mathcal{T}^{-1/4}). \tag{18}$$

This result can be obtained by using Theorem 4.1. Thus, $\lim_{\mathcal{T}\to\infty} \frac{R(\mathcal{T})}{\mathcal{T}} = 0$. More details are proided in Appendix A. The findings demonstrate that BSO and T-BSO algorithms will achieve convergence throughout the training phase, provided that gradients remain bounded. The convergence analysis is crucial as it guarantees the stability and effectiveness of the optimization algorithms, ensuring that they will eventually reach an optimal or near-optimal solution as the training progresses.

**Complexity Analysis** We analyze the memory requirements for weight storage and computation state during training, as summarized in Table 1. Let $n$ represent the average number of neurons per layer. BPTT with SG maintains the entire computational graph to enable backpropagation of gradients through time. Consequently, BPTT requires time-dependent memory for gradient computation and incurs an

$O(n^2L)$ storage cost for latent weights; although OTTT is time-independent, it still needs latent weights to the same extent as BPTT. In contrast, BSO and T-BSO significantly reduce weight storage requirements by eliminating the need for full-precision latent weights. Building on this approach, BSO achieves time-independent memory costs by leveraging an online training framework. Additionally, T-BSO incorporates neglect variables into the optimizer to capture temporal dynamics ($T \ll n^2$).

---

**Algorithm 1** BSO and T-BSO for optimization.
---
1: **Input:** $T$ : Number of time steps; $\epsilon$ : Small constant to prevent zero threshold; $\beta_1, \beta_2$ : Decay rates for momentum; $\gamma$ : Threshold; $\mathcal{L}$ : Loss function; $W^l$ : Binary network weights at layer $l$; $x$ : Input data; $y$ : Ground truth labels
2: **Output:** Optimized binary weights $W^l$
3: $M^l \leftarrow 0$ Initialize $1^{st}$ moment vector
4: $v^l \leftarrow 0$ Initialize temporal $2^{nd}$ moment vector
5: **while** not converged **do**
6:   **for** $t \leftarrow 1$ to $T$ **do**
7:     **for** $l \leftarrow 1$ to $N$ **do**
8:       $u^l[t] \leftarrow \lambda(u^l[t-1] - V_{th}s^l[t-1]) + W^l s^{l-1}[t]$
9:       $s^l[t] \leftarrow H(u^l[t] - V_{th})$
10:       $\hat{a}^l[t] \leftarrow \lambda\hat{a}^l[t-1] + s^l[t]$
11:     **end for**
12:     **for** $l \leftarrow N$ downto 1 **do**
13:       $G[t] \leftarrow \nabla_{W^l}L[t]$
14:       $M^l \leftarrow \beta_1 \cdot M^l + (1 - \beta_1) \cdot G[t]$
15:       **if** T-BSO **then**
16:         $v^l[t] \leftarrow \beta_2 \cdot v^l[t] + (1 - \beta_2) \cdot \text{mean}(G^2[t])$
17:       **end if**
18:       **if** T-BSO **then**
19:         $W^l \leftarrow -\text{sign}(W^l \odot M^l - \gamma\sqrt{v^l[t] + \epsilon}) \odot W^l$
20:       **else**
21:         $W^l \leftarrow -\text{sign}(W^l \odot M^l - \gamma) \odot W^l$
22:       **end if**
23:     **end for**
24:   **end for**
25: **end while**

---

## 5. Experiments

In this section, we first present experimental details, including the utilized datasets, architectures, and settings. Second, we compare our methods with existing online training and BSNN approaches to evaluate their effectiveness. Thirdly, we conduct ablation studies to assess the efficiency improvements of BSO and T-BSO during the training process, as well as the performance advantages of T-BSO over BSO. Finally, we analyze the hyperparameter and scalability study of our more advanced T-BSO.

*Table 2.* Comparisons of BSO and T-BSO with related work, focusing on the online training and BSNN methods in the SNN domain.

| DATASET | METHOD | ARCHITECTURE | OLINE TRAINING | BINARY WEIGHT | TIME STEP | MODEL SIZE (MB) | ACC. (%) |
|---|---|---|---|---|---|---|---|
| CIFAR-10 | FP-SNN | VGG-11 | ✗ | ✗ | 6 | 36.91 BASE | 93.23 BASE |
| | OTTT (XIAO ET AL., 2022) | VGG-11 | ✔ | ✗ | 6 | 36.91 (1.0×) | 93.73 (+0.5) |
| | NDOT (JIANG ET AL.) | VGG-11 | ✔ | ✗ | 6 | 36.91 (1.0×) | 94.90 (+1.7) |
| | CBP (YOO & JEONG, 2023) | VGG-16 | ✗ | ✔ | 32 | 15.10 (2.4×) | 91.51 (-1.7) |
| | Q-SNN (WEI ET AL., 2024) | VGG-11 | ✗ | ✔ | 6 | 1.20 (30.8×) | 93.20 (+0.03) |
| | BSO | VGG-11 | ✔ | ✔ | 6 | 1.20 (30.8×) | 92.98 (-0.3) |
| | | | | | 4 | | 93.45 (+0.2) |
| | | | | | 2 | | 93.30 (+0.1) |
| | T-BSO | VGG-11 | ✔ | ✔ | 6 | 1.20 (30.8×) | 94.70 (+1.5) |
| | | | | | 4 | | 94.86 (+1.6) |
| | | | | | 2 | | 94.32 (+1.1) |
| CIFAR-100 | FP-SNN | VGG-11 | ✗ | ✗ | 6 | 37.10 BASE | 71.15 BASE |
| | OTTT (XIAO ET AL., 2022) | VGG-11 | ✔ | ✗ | 6 | 37.10 (1.0×) | 71.11 (-0.04) |
| | NDOT (JIANG ET AL.) | VGG-11 | ✔ | ✗ | 6 | 37.10 (1.0×) | 76.61 (+5.5) |
| | CBP (YOO & JEONG, 2023) | VGG-16 | ✗ | ✔ | 32 | 16.60 (2.2×) | 66.53 (-4.6) |
| | Q-SNN (WEI ET AL., 2024) | VGG-11 | ✗ | ✔ | 6 | 1.39 (26.7×) | 73.48 (+2.3) |
| | BSO | VGG-11 | ✔ | ✔ | 6 | 1.39 (26.7×) | 72.57 (+1.4) |
| | | | | | 4 | | 72.34 (+1.2) |
| | | | | | 2 | | 72.15 (+1.0) |
| | T-BSO | VGG-11 | ✔ | ✔ | 6 | 1.39 (26.7×) | 74.27 (+3.1) |
| | | | | | 4 | | 73.82 (+2.7) |
| | | | | | 2 | | 73.40 (+2.3) |
| IMAGENET | FP-SNN | RESNET-18 | ✗ | ✗ | 4 | 87.19 BASE | 63.18 BASE |
| | SLTT (MENG ET AL., 2023) | RESNET-34 | ✔ | ✗ | 6 | 85.15 (1.0×) | 66.19 (+3.0) |
| | OTTT (XIAO ET AL., 2022) | RESNET-34 | ✔ | ✗ | 6 | 87.19 (1.0×) | 64.16 (+1.0) |
| | CBP (YOO & JEONG, 2023) | RESNET-18 | ✗ | ✔ | 4 | 4.22 (20.7×) | 54.34 (-8.8) |
| | T-BSO | RESNET-18 | ✔ | ✔ | 6 | 4.22 (20.7×) | 58.86 (-4.3) |
| | | | | | 4 | | 57.76 (-5.4) |
| DVS-CIFAR10 | FP-SNN | VGG-11 | ✗ | ✗ | 10 | 36.91 BASE | 73.90 BASE |
| | SLTT (MENG ET AL., 2023) | VGG-11 | ✔ | ✗ | 10 | 34.32 (1.1×) | 82.20 (+8.3) |
| | NDOT (JIANG ET AL.) | VGG-11 | ✔ | ✗ | 10 | 36.91 (1.0×) | 77.50 (+3.6) |
| | CBP (YOO & JEONG, 2023) | 16CONV1FC | ✗ | ✔ | 16 | 1.33 (27.8×) | 74.70 (+0.8) |
| | Q-SNN (WEI ET AL., 2024) | VGG-11 | ✗ | ✔ | 10 | 1.20 (30.8×) | 79.10 (+5.2) |
| | BSO | VGG-11 | ✔ | ✔ | 10 | 1.20 (30.8×) | 80.60 (+6.7) |
| | T-BSO | VGG-11 | ✔ | ✔ | 10 | 1.20 (30.8×) | 81.00 (+7.1) |

## 5.1. Implementation Details

We validate our BSO and T-BSO on image classification tasks, including both static image datasets like CIFAR-10 (Krizhevsky et al., 2009), CIFAR100 (Krizhevsky et al., 2009), ImageNet (Deng et al., 2009), as well as the neuromorphic dataset DVS-CIFAR10 (Li et al., 2017). Regarding the architecture, we use VGG (64C3-128C3-AP2-256C3-256C3-AP2-512C3-512C3-AP2-512C3-512C3-GAP-FC) and ResNet-18 (Liu et al., 2018), which are commonly used in the SNN domain (Xiao et al., 2022; Jiang et al.). Consistent with prior work (Rastegari et al., 2016; Qin et al., 2020), we maintain full-precision weights in the first and final layers. As for the training configuration, we apply SGD with a cosine annealing learning rate schedule. The

initial learning rate is set to 0.1 and decays to 0 throughout the training process. Moreover, we set the threshold $V_{th}$ and the decay factor $\lambda$ in the LIF to 1 and 0.5. For the hyperparameters of our BSO and T-BSO, we set $\gamma$ to $5 \times 10^{-7}$, while $\beta_1$ and $\beta_2$ to $1 - 10^{-3}$ and $1 - 10^{-5}$. We provide the hyperparameter analysis for them in Sec. 5.4. When experimenting on the ImageNet, we first pre-train the model using single time steps for 100 epochs, then fine-tune with T-BSO using 4-6 time steps for 30 epochs to accelerate training. Additional details are available in Appendix B.

## 5.2. Comparison of Performance

We evaluate our BSO and T-BSO methods on static and neuromorphic datasets and compare with related work, mainly

focusing on online training methods (Xiao et al., 2022; Meng et al., 2023; Jiang et al.) and BSNN methods (Wei et al., 2024; Yoo & Jeong, 2023).

As shown in Table 2, on the CIFAR-10 dataset, T-BSO achieves 94.70%, 94.86%, and 94.32% accuracy with 6, 4, and 2 time steps, respectively. With a reduction in model size by 30.76×, our approach achieves performance comparable to existing online training methods and even surpasses OTTT by 0.97%. For CIFAR-100, T-BSO also demonstrates superior performance, achieving the accuracy of 74.27%, 73.82% and 73.40% with 6, 4 and 2 time steps. Compared to BPTT-based BSNN methods like Q-SNN, T-BSO achieves performance improvements of 0.79%, while requiring less training overhead. On the ImageNet, with the same time steps, T-BSO achieves an accuracy of 57.76%, which is 3.42% higher than the BPTT-based CBP method. Although there is still a gap between T-BSO and methods like SLTT and OTTT on this dataset, it's worth noting that T-BSO achieves these competitive results with a model size of only 4.22MB (reduced by 20.18×, 20.66×). On the neuromorphic DVS-CIFAR10, T-BSO achieves 81.00% accuracy with 10 time steps, outperforming both existing online training and BSNN methods. These results demonstrate that our proposed BSO and T-BSO methods maintain the efficiency advantages of BSNNs in both forward and backward propagation, while achieving superior performance.

## 5.3. Ablation Study

This part presents ablation studies on BSO and T-BSO. First, we compare the memory overhead during training with the widely used BPTT learning method in BSNNs, demonstrating the efficiency of our BSO and T-BSO in backpropagation. Second, we validate the effectiveness of the temporal-aware mechanism by analyzing T-BSO's performance improvement over BSO.

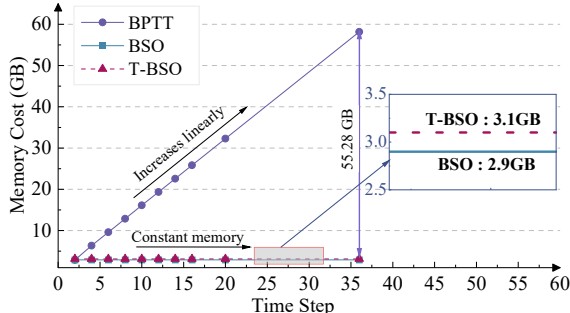

*Figure 4.* Memory cost comparison between the widely used BPTT algorithm in BSNN and our BSO/T-BSO across varying time steps.

**Training cost comparison for three BSNN learning methods (i.e., BPTT vs. BSO vs. T-BSO).** BSO and T-BSO eliminate backpropagation through time, achieving

time-invariant memory consumption and avoiding BPTT's quadratic memory overhead. We validate this through training memory measurements on CIFAR-10 under identical configurations (backbone, batch size, etc.). As shown in Figure 4, BPTT algorithms commonly used in existing BSNNs increase memory usage with time steps, whereas BSO and T-BSO maintain constant memory. Specifically, when the time step grows from 1 to 35, the memory cost of BPTT increases from 2.86GB to 58.18GB (20.3×), while our BSO and T-BSO both maintain constant memory usage, 2.9GB and 3.1GB, respectively. The minor overhead in T-BSO arises from its temporal-aware module. These results demonstrate that our BSO and T-BSO algorithm significantly reduce memory requirements during training.

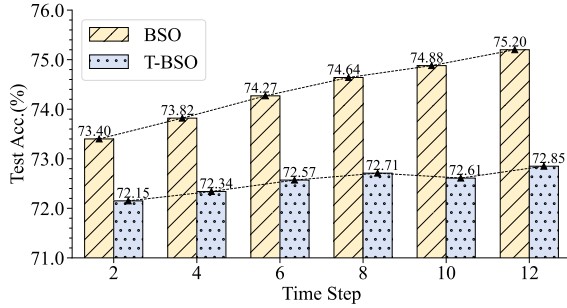

*Figure 5.* Performance comparison between our BSO and T-BSO under different time steps. T-BSO consistently outperforms BSO across time steps and exhibits a larger increase over time steps.

**Comparative analysis of our BSO and T-BSO algorithms.** In T-BSO, we introduce the time-aware threshold to consider the time information when updating the weights. We evaluate the accuracy of BSO and T-BSO at different time steps using experiments on the CIFAR-100 dataset and the VGG-11 network. The experimental results are shown in Figure 5. It can be seen that both BSO and T-BSO show improved performance with the increase of time steps. In addition, due to the better utilization of temporal information by the temporal-aware threshold, T-BSO always outperforms BSO in each time step, and the performance increases significantly with the time step.

## 5.4. Hyperparameter and Scalability Study of T-BSO

As discussed above, we demonstrate the training efficiency of BSO and T-BSO, as well as the superior performance of T-BSO. In this part, we conduct a more in-depth analysis of our advanced T-BSO learning algorithm, including the hyperparameter study and the scalability to non-vision tasks.

**Hyperparameter study of T-BSO.** Given that T-BSO introduces new hyperparameters, we analyze its sensitivity to parameter variations, including threshold $\gamma$ and the

Table 3. Effect of the threshold $\gamma$ in T-BSO on performance.

| $\gamma$ ($\times 10^{-5}$) | 0.005 | 0.02 | 0.05 | 0.08 |
|---|---|---|---|---|
| ACCURACY (%) | 73.75 | 73.33 | 73.40 | 73.24 |
| $\gamma$ ($\times 10^{-5}$) | 0.1 | 0.15 | 0.2 | 5.0 |
| ACCURACY (%) | 73.38 | 73.34 | 73.41 | 72.83 |

Table 4. Effect of decay rates for momentum in T-BSO, i.e., $\beta_1$ and $\beta_2$, on performance.

| $\gamma$ | $1 - \beta_1$ | $1 - \beta_2$ | ACCURACY |
|---|---|---|---|
| | | $1.5 \times 10^{-6}$ | 73.22 |
| | $1.5 \times 10^{-4}$ | $1.0 \times 10^{-5}$ | 73.31 |
| $5.0 \times 10^{-7}$ | | $1.5 \times 10^{-5}$ | 73.29 |
| | | $1.5 \times 10^{-6}$ | 73.17 |
| | $1.0 \times 10^{-3}$ | $1.0 \times 10^{-5}$ | 73.40 |
| | | $1.5 \times 10^{-5}$ | 73.34 |

decay rates for momentum ($\beta_1$ and $\beta_2$). Experiments are performed using VGG-11 architecture on CIFAR-100 with T=2 time steps. The analysis of $\gamma$ is presented in Table 3. The threshold parameter $\gamma$ maintains stable performance across a range from $5.0 \times 10^{-8}$ to $5.0 \times 10^{-5}$, with accuracy varying only between 72.83% and 73.41%. This stability demonstrates low sensitivity to threshold selection. The analysis of decay rates for momentum is shown in Table 4. Despite the wide range of decay rates settings, performance variations remain minimal. Optimal performance is achieved with smaller decay rates ($1 - \beta_1 = 1.5 \times 10^{-4}$, $1 - \beta_2 = 1.5 \times 10^{-5}$), yielding an accuracy of 73.29%. These results demonstrate that while T-BSO's design introduces several hyperparameters, it exhibits low sensitivity to hyperparameters. Such robustness is beneficial, as it guarantees consistent performance across a variety of settings.

**Scalability Study of T-BSO.** Existing online training methods have been primarily validated on image tasks, leaving their scalability to other domains unexplored. To address this limitation, we evaluate T-BSO's effectiveness on audio tasks by conducting experiments on the Google Speech Command (GSC) dataset (Warden, 2018). We evaluate T-BSO by employing the VGG-11 backbone with $T = 4$, as shown in Table 5. Our approach is benchmarked against existing SNN methods, specifically ATIF (Castagnetti et al., 2023) and STS-T (Wang et al., 2023). Our exploration on the GSC dataset demonstrates T-BSO's superior performance. T-BSO achieves the highest accuracy of 96.12% while being the only method supporting online training. Additionally, T-BSO operates with 1-bit quantization and maintains the model size of 1.26MB. These findings show that our approach extends well to non-visual tasks, highlighting its applicability beyond image-based uses.

Table 5. Performance comparison of T-BSO on the GSC dataset.

| METHOD | ONLINE TRAINING | BINARY WEIGHT | TIME STEP | MODEL SIZE (MB) | ACC. (%) |
|---|---|---|---|---|---|
| ATIF | ✗ | ✗ | 4 | 44.74 | 94.31 |
| STS-T | ✗ | ✗ | 4 | 6.24 | 95.18 |
| T-BSO | ✔ | ✔ | 4 | 1.26 | 96.12 |

## 6. Conclusion

This paper introduces BSO, a novel online optimization algorithm designed specifically for BSNNs. Unlike existing training methods, BSO eliminates the need for such auxiliary storage by directly updating binary weights through a flip-based mechanism guided by gradient momentum. This design preserves the efficiency advantages of BSNNs in both forward and backward processes. To further improve performance, the paper proposes T-BSO, a temporal-aware extension of BSO that incorporates second-order gradient moments to dynamically adjust flipping thresholds across time steps. This enhancement captures the intrinsic temporal dynamics of spiking activity and leads to more stable and effective optimization. Theoretical analysis establishes regret bounds for both BSO and T-BSO, confirming convergence under standard assumptions. Experiments demonstrate that BSO and T-BSO achieve competitive accuracy while significantly reducing training overhead, representing a substantial advance toward efficient and scalable BSNN training.

## Acknowledgements

This work is supported in part by the National Natural Science Foundation of China (No. U2333211, U20B2063 and 62220106008), in part by the Project of Sichuan Engineering Technology Research Center for Civil Aviation Flight Technology and Flight Safety (No. GY2024-27D), in part by the Open Research Fund of the State Key Laboratory of Brain-Machine Intelligence, Zhejiang University (Grant No.BMI2400020), in part by the Shenzhen Science and Technology Program (Shenzhen Key Laboratory, Grant No. ZDSYS20230626091302006), in part by the Shenzhen Science and Technology Research Fund (Fundamental Research Key Project, Grant No. JCYJ20220818103001002) and in part by the Program for Guangdong Introducing Innovative and Enterpreneurial Teams, Grant No. 2023ZT10X044.

## Impact Statement

This paper presents work whose goal is to advance the field of Machine Learning. There are many potential societal consequences of our work, none which we feel must be specifically highlighted here.

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

# A. Convergence Proof

In this section, we prove Theorem 4.1.

**Definition A.1.** A function $f : R^d \to R$ is convex if for any $x, y \in R$, for all $\lambda \in [0, 1]$,

$$\lambda f(x) + (1 - \lambda)f(x) \geq f(\lambda x + (1 - \lambda)y). \tag{19}$$

Also, notice that a convex function can be lower bounded by a hyperplane at its tangent.

**Lemma A.2.** *If a function $f : R^d \to R$ is convex, then for all $x, y \in R^d$,*

$$f(y) \geq f(x) + \nabla f(x)^T (y - x). \tag{20}$$

The above lemma can be used to upper bound the regret and our proof for the main theorem is constructed by substituting the hyperplane with the BSO update rules.

Consider an online optimization scenario, where at each round $k \in 1, 2, \dots, \mathcal{T}$, an algorithm predicts the parameter $w_k$ and incurs a loss based on a convex cost function $f_k(w)$. The sequence $\{f_k\}_{k=1}^{\mathcal{T}}$ is arbitrary and unknown in advance.

**Definition A.3.** The regret $R(\mathcal{T})$ of an online algorithm over $\mathcal{T}$ rounds is defined as:

$$R(\mathcal{T}) = \sum_{k=1}^{\mathcal{T}} f_k(W_k) - \min_{W^* \in \mathcal{X}} \sum_{k=1}^{\mathcal{T}} f_k(W^*), \tag{21}$$

where $\mathcal{X}$ is the feasible set of parameters.

Our objective is to analyze the regret of BSO and demonstrate that it achieves sublinear regret, implying that the average regret per time step diminishes as $\mathcal{T}$ increases.

**Assumption A.4** (Convexity). Each loss function $f_k(W)$ is convex with respect to $W$ for all $k = 1, 2, \dots, \mathcal{T}$.

**Assumption A.5** (Bounded Gradients). There exists a constant $G > 0$ such that for all $k$,

$$\|\nabla_W \mathcal{L}_k(W)\| \leq G.$$

**Theorem A.6.** *Let $f_k$ be a function with bounded gradients such that $|\nabla f_k(w)|_2 \leq G$ and $|\nabla f_w(w)|_\infty \leq G_\infty$ for all $w \in \mathbb{R}^d$. Furthermore, assume that any $w$ generated by BSO remains bounded. And the $\gamma$ and $\beta$ decay by $\sqrt{k}$. Then BSO achieves a regret bound of $R(\mathcal{T}) = O(\sqrt{\mathcal{T}})$.*

*Proof.* First, for each iteration k and flipped position i:

$$W_{k,i} M_{k,i} - \gamma \geq 0, \tag{22}$$

the inner product decomposes as:

$$\langle M_k, W_k - W^* \rangle = \sum_{i=1}^{d} M_{k,i}(W_{k,i} - W_i^*), \tag{23}$$

For each dimension i, since $W_{k,i}, W^i \in -1, 1$:

$$|W_{k,i} - W^i| \leq |W_{k,i}| + |W_i^*| \leq 2. \tag{24}$$

For each iteration k:

$$\langle M_k, W_k - W^* \rangle \leq |M_k|_\infty |W_k - W^*|_1 + 2\gamma |i : W_{k+1,i} \neq W_{k,i}|. \tag{25}$$

Consider the recursion:

$$|M_k|_\infty \leq \frac{\beta}{\sqrt{k}} G + (1 - \frac{\beta}{\sqrt{k}})|M_{k-1}|_\infty. \tag{26}$$

By induction, this leads to:

$$|M_k|_\infty = O(\frac{1}{\sqrt{k}}). \tag{27}$$

Therefore:

$$\sum_{k=1}^{\mathcal{T}} |M_k|_\infty \leq \sum_{k=1}^{\mathcal{T}} O(\frac{1}{\sqrt{k}}) = O(\sqrt{\mathcal{T}}). \tag{28}$$

Finally:

$$R_T \leq 2\sum_{k=1}^{\mathcal{T}} \gamma_k + 2\sum_{k=1}^{\mathcal{T}} |M_k|_\infty = O(\sqrt{\mathcal{T}}). \tag{29}$$

$\square$

**Theorem A.7.** *Let $f_k$ be a function with bounded gradients such that $|\nabla f_k(w)|_2 \leq G$ and $|\nabla f_k(w)|_\infty \leq G_\infty$ for all $w \in \mathbb{R}^d$. Furthermore, assume that any $w$ generated by BSO remains bounded. When $\gamma_k$ and $\beta_{1,k}, \beta_{2,k}$ decay by $1/\sqrt{k}$, the algorithm achieves a regret bound of $R(\mathcal{T}) = O(T^{3/4})$.*

*Proof.* First, we establish the moment bounds. For the first-order moment:

$$M_k = \frac{\beta_1}{\sqrt{k}} M_{k-1} + (1 - \frac{\beta_1}{\sqrt{k}}) G_k. \tag{30}$$

Through recursion, we obtain:

$$|M_k|_\infty = O(1/\sqrt{k}). \tag{31}$$

For the second-order moment:

$$v_k = \frac{\beta_2}{\sqrt{k}} \cdot v_{k-1} + (1 - \frac{\beta_2}{\sqrt{k}}) \cdot G_k^2. \tag{32}$$

Similarly:

$$|v_k|_\infty = O(1/\sqrt{k}). \tag{33}$$

For the critical ratio, we derive:

$$|M_k/\sqrt{v_k + \epsilon}|_\infty = |M_k|_\infty/\sqrt{|v_k|_\infty} = O(1/\sqrt{k})/\sqrt{O(1/\sqrt{k})} = O(k^{-1/4}). \tag{34}$$

At each iteration k and flipped position i:

$$W_{k,i} M_{k,i}/\sqrt{v_{k,i}} - \gamma_k \geq 0. \tag{35}$$

The inner product decomposes as:

$$\langle M_k, W_k - W^* \rangle = \sum_{i=1}^{d} M_{k,i}(W_{k,i} - W_i^*). \tag{36}$$

For each dimension i, since $W_{k,i}, W^i \in -1, 1$:

$$|Wk, i - W_{,i}| \leq 2. \tag{37}$$

Therefore, at each iteration k:

$$\langle M_k, W_k - W^* \rangle \leq |M_k|_\infty |W_k - W^*|_1 + 2\gamma_k |i : W_{k+1,i} \neq W_{k,i}|. \tag{38}$$

From the moment bounds:

$$\sum_{k=1}^{\mathcal{T}} |M_k/\sqrt{v_k + \epsilon}|_\infty \leq \sum_{k=1}^{\mathcal{T}} O(k^{-1/4}) = O(\mathcal{T}^{3/4}). \tag{39}$$

Finally:

$$R_{\mathcal{T}} \leq 2\sum_{k=1}^{\mathcal{T}} \gamma_k + 2\sum_{k=1}^{\mathcal{T}} |M_k^l/\sqrt{v_k^l + \epsilon}|_\infty = O(\mathcal{T}^{3/4}). \tag{40}$$

$\square$

*Table 6.* Training hyperparameters about BSO

| Dataset | Epoch | Batch size | $\gamma$ | $\beta$ | LR |
|---|---|---|---|---|---|
| CIFAR-10 | 400 | 128 | 5e-7 | $1-10^{-3}$ | 0.1 |
| CIFAR-100 | 400 | 128 | 5e-7 | $1-10^{-3}$ | 0.1 |
| DVS-CIFAR10 | 400 | 128 | 1e-6 | $1-10^{-3}$ | 0.1 |

*Table 7.* Training hyperparameters about T-BSO

| Dataset | Epoch | Batch size | $\gamma$ | $\beta_1$ | $\beta_2$ | LR |
|---|---|---|---|---|---|---|
| CIFAR-10 | 400 | 128 | 5e-7 | $1-10^{-3}$ | $1-10^{-5}$ | 0.1 |
| CIFAR-100 | 400 | 128 | 5e-7 | $1-10^{-3}$ | $1-10^{-5}$ | 0.1 |
| ImageNet(pre-train) | 100 | 64 | 1e-6 | $1-10^{-3}$ | $1-10^{-5}$ | 0.1 |
| ImageNet(finetune) | 30 | 64 | 1e-11 | $1-10^{-8}$ | $1-10^{-5}$ | 0.1 |
| DVS-CIFAR10 | 400 | 128 | 1e-6 | $1-10^{-3}$ | $1-10^{-5}$ | 0.1 |

# B. Implementation Details

### B.1. Datasets

We conduct experiments on CIFAR-10 (Krizhevsky et al., 2009), CIFAR-100 (Krizhevsky et al., 2009), ImageNet (Deng et al., 2009), CIFAR10-DVS (Li et al., 2017).

**CIFAR-10** CIFAR-10 is a dataset consisting of color images across 10 object categories, with 50,000 training samples and 10,000 testing samples. Each image is $32 \times 32$ pixels with three color channels. We preprocess the data by normalizing the inputs using the global mean and standard deviation and apply data augmentation techniques including random cropping, horizontal flipping, and cutout (DeVries, 2017). At each time step, the input to the first layer of the SNNs corresponds directly to the pixel values, which can be interpreted as a real-valued input current.

**CIFAR-100** CIFAR-100 is a dataset similar to CIFAR-10 but contains 100 object categories instead of 10. It includes 50,000 training samples and 10,000 testing samples, with the same preprocessing applied as in CIFAR-10. Both CIFAR-10 and CIFAR-100 are distributed under the MIT License.

**ImageNet** ImageNet-1K is a large-scale dataset consisting of color images across 1,000 object categories, with 1,281,167 training samples and 50,000 validation images. We apply standard preprocessing techniques, where training images are randomly resized and cropped to $224 \times 224$, followed by normalization after random horizontal flipping for data augmentation. For testing images, they are resized to $256 \times 256$, center-cropped to $224 \times 224$, and then normalized. At each time step, the inputs are converted into a real-valued input current.

**DVS-CIFAR10** The DVS-CIFAR10 dataset is a neuromorphic version of the CIFAR-10 dataset, generated using a Dynamic Vision Sensor (DVS). It contains 10,000 samples, one-sixth of the original CIFAR-10, and consists of spike trains with two channels corresponding to ON- and OFF-event spikes. The pixel resolution is expanded to $128 \times 128$. We split the dataset into 9,000 training samples and 1,000 testing samples. For data preprocessing, we reduce the time resolution by accumulating spike events (Fang et al., 2021) into 10 time steps and lower the spatial resolution to $48 \times 48$ through interpolation. We apply random cropping augmentation to the input data, similar to CIFAR-10, and normalize the inputs based on the global mean and standard deviation across all time steps.

### B.2. Settings

For our proposed BSO and T-BSO algorithm, we apply the NDOT and OTTT for online training framework. The training hyper-parameters are detailed in Table 6 and Table 7. Our BSO and T-BSO are tested with online update strategies, which updates the parameters in real-time at each time step to fully explore the potential of online training for BSNNs. For a fair comparison, we implement our Q-SNN (Wei et al., 2024) based on the VGG-11 architecture. For the GSC dataset, we employ a VGG-11 architecture with $T = 4$. The training hyperparameters are set as:$\gamma = 5 \times 10^{-7}$, $\beta_1 = 0.001$, $\beta_2 = 1 \times 10^{-5}$. The GSC data are processed as follows: raw audio signals are converted to Mel-spectrograms, which are then resized to 32×32 pixels and normalized with a mean of 0.5 and a standard deviation of 0.5.

