# OpenReview forum: "BSO: Binary Spiking Online Optimization Algorithm"
_ICML.cc/2025/Conference — ICML 2025 poster_

### Official Review · Reviewer_CwUQ · 2025-03-13

**Overall Recommendation:** 4

**Summary:**

The paper introduces the Binary Spiking Online Optimization (BSO) algorithm, an approach to training BSNNs that reduces training memory overhead while maintaining computational efficiency. The basic BSO eliminates latent weight storage and uses momentum-based gradient accumulation to generate weight-flipping signals, while the temporal-aware T-BSO variant captures gradient information across time steps with an adaptive threshold mechanism. Extensive experimental validation validates the effectiveness of BSO and T-BSO.

**Claims And Evidence:**

Yes, most of the claims are supported by appropriate evidence.

**Essential References Not Discussed:**

No.

**Experimental Designs Or Analyses:**

I examined the experimental designs and analyses presented in the paper, focusing on the validation of the BSO and T-BSO algorithms across different datasets and comparisons with baseline methods. (1) The authors compare BSO and T-BSO against both online training methods (OTTT, NDOT, SLTT) and BSNN methods (Q-SNN, CBP). (2) The training paradigms are analyzed using the FF ratio and C2I ratio metrics to investigate the optimization stability.

**Methods And Evaluation Criteria:**

The proposed methods and evaluation criteria in the BSO paper are generally well-aligned with the problem they aim to address.

**Other Comments Or Suggestions:**

No.

**Other Strengths And Weaknesses:**

Strengths:
1. By eliminating latent weights and making memory requirements independent of time steps, the BSO and T-BSO provide a meaningful solution for resource-constrained applications.
2. The theoretical analysis is comprehensive. The formal regret bounds provide mathematical guarantees for convergence.
3. The ablation studies illustrate the effectiveness of the methods, particularly the role of momentum in stabilizing training for binary networks.

Weaknesses & Questions:
1.The paper lacks analysis of how the proposed methods would perform with more complex network architectures beyond standard VGG and ResNet models, such as transformer-based architectures or recurrent networks.
2.Do BSO and T-BSO lead to reductions in operations (OPs/SOPs) and energy consumption? These metrics are crucial for edge deployment scenarios.
3.The authors only validated the proposed methods on simple image classification tasks. Can they be extended to other applications?
4.There's no analysis of BSO's convergence behavior under noisy data, which would be valuable for real-world applications where data quality varies.
5.There is significant room for improvement in the writing of the paper.

**Questions For Authors:**

See Weaknesses & Questions.

**Relation To Broader Scientific Literature:**

The BSO paper makes contributions to several domains, such as SNNs, online learning, and efficient computing. By integrating these research domains, BSO addresses the contradiction between BSNNs' inherent efficiency and their memory-intensive training processes.

**Theoretical Claims:**

I examined the theoretical claims and proofs presented in the paper, focusing on the convergence guarantees for both BSO and T-BSO algorithms. The theoretical analysis is mathematically sound.

---

> ### Author Rebuttal · Authors · 2025-03-31
>
> **Response to Q1:"The paper lacks analysis of how the proposed methods would perform with more complex network architectures beyond standard VGG and ResNet models, such as transformer-based architectures or recurrent networks."**
>
> We agree that evaluating BSO and T-BSO on more advanced architectures would strengthen our claims. While we focus on VGG and ResNet for their interpretability and popularity in the BSNN literature, our method is **model-agnostic**. This makes it naturally extendable to hierarchical and attention-based architectures such as spiking Transformers or recurrent SNNs. We have discussed the Spike-Former results in our response to Reviewer Tbrf (Response to W2) and kindly refer you to that section for transformer-based results. In a similar vein, we explored the FeedBack (recurrent) SNN [1] and employed VGG11 with T-BSO, training for 300 epochs on CIFAR-10, achieving an accuracy of 92.43%. This demonstrates the versatility of our method and its ability to extend naturally to **different network architectures**.
>
> **Response to Q2:"Do BSO and T-BSO lead to reductions in operations (OPs/SOPs) and energy consumption?"**
>
> As mentioned in our response to Reviewer Q2xs (Response to W4), we have explained that BSO and T-BSO result in reduced operations and energy consumption due to their lower firing rates. We kindly refer you to that section for a detailed explanation.
>
> **Response to Q3:"The authors only validated the proposed methods on simple image classification tasks. Can they be extended to other applications"**
>
> Our initial experiments focused on classification as a standard benchmark for spiking models. However, the proposed method is task-agnostic and can be integrated into other applications. As stated in our response to Reviewer Q2xs (Response to W2), we conducted experiments on the GSC dataset, which demonstrates the effectiveness of our method across different tasks.
>
> **Response to Q4:"There's no analysis of BSO's convergence behavior under noisy data, which would be valuable for real-world applications where data quality varies."**
>
> This is an important consideration for real-world deployment. The update mechanism of BSO, which relies on the momentum of the gradient exceeding a threshold to induce a sign flip, inherently offers robustness against noise in the data. This is due to the thresholding operation, which acts as a filtering mechanism, allowing only **significant gradient information** to influence the weight updates. As a result, smaller fluctuations in the gradient caused by noise are less likely to trigger updates, making the algorithm less sensitive to noisy data. The training methods of BSO and T-BSO are not only resilient to gradient noise fluctuations but also to the noise in the input data. To validate this, we conduct experiments using the CIFAR-10, CIFAR-100, and VGG-11 architectures, where three different types of Gaussian noise are added to the input images during testing. The performance of BSO and T-BSO is then compared in terms of accuracy. The experimental results demonstrate that both BSO and T-BSO exhibit a certain level of noise robustness, with T-BSO outperforming BSO under identical noise conditions.
>
> | Dataset   | Algorithm | Base  | $(\mu=0,\sigma=0.1)$ | $(\mu=0,\sigma=0.4)$ | $(\mu=0.3,\sigma=0.1)$ |
> |----------|-------|-------|----------------------|----------------------|------------------------|
> | **CIFAR-100** | BSO   | 68.73 | 65.82 (-2.91)        | 46.42 (-22.31)       | 64.95 (-3.78)          |
> |          | T-BSO | 74.17 | 70.34 (-3.83)        | 51.21 (-22.96)       | 69.45 (-4.72)          |
> | **CIFAR-10**  | BSO   | 93.30 | 91.02 (-2.28)        | 80.61 (-12.69)       | 92.75 (-0.55)          |
> |          | T-BSO | 94.32 | 93.01 (-1.31)        | 85.47 (-8.85)        | 93.12 (-1.20)          |
>
> **Response to Q5:"There is significant room for improvement in the writing of the paper."**
>
> Thank you for your candid feedback. We sincerely apologize for our poor writing. We will carefully revise the manuscript to improve clarity, coherence, and presentation. This includes refining the descriptions of our methods, improving figure quality throughout the paper.
>
> [1] Xiao, Mingqing, et al. "Training feedback spiking neural networks by implicit differentiation on the equilibrium state." Advances in neural information processing systems 34 (2021): 14516-14528.

---

### Official Review · Reviewer_AqeW · 2025-03-13

**Overall Recommendation:** 4

**Summary:**

This paper introduces the Binary Spiking Online Optimization (BSO) algorithm, designed to reduce memory overhead in training Binary Spiking Neural Networks (BSNNs) by eliminating latent weight storage and making memory requirements time-independent. It also presents T-BSO, a temporal-aware variant that adjusts thresholds dynamically using gradient information across time steps for improved optimization.

**Claims And Evidence:**

Yes

**Essential References Not Discussed:**

No.

**Experimental Designs Or Analyses:**

Yes

**Methods And Evaluation Criteria:**

Yes

**Other Comments Or Suggestions:**

No

**Other Strengths And Weaknesses:**

Strengths:
BSO reduces memory overhead by eliminating latent weight storage, making it ideal for resource-constrained environments; Experimental results show better accuracy and efficiency than existing methods.
Weakness:
T-BSO introduces more computational overhead due to additional gradient computations.
The algorithms require careful tuning of hyperparameters, which can be time-consuming.

**Questions For Authors:**

1. The author mentioned that BSO is based on a direct learning method. Why not consider the conversion-based method, but choose the direct training method?
2. The author points out in the introduction, 'leverage the substantial efficiency advantages in BSNN training'. Specifically, how does the BSO algorithm demonstrate its advantages in BSNN?
3. Does the introduction of adaptive thresholds in the T-BSO version of BSO affect the neural dynamics of the online learning framework?

**Relation To Broader Scientific Literature:**

This article proposes the BSO algorithm and its variant T-BSO, further promoting edge intelligent computing.

**Theoretical Claims:**

Yes, I have checked the proof and theorems.

---

> ### Author Rebuttal · Authors · 2025-03-31
>
> **Response to W1:"T-BSO introduces more computational overhead due to additional gradient computations."**
>
> As noted in our Response to Reviewer Q2xs (Response to W1), we have discussed that T-BSO incurs additional computational overhead due to the necessity for extra gradient computations. We kindly direct you to that section for a comprehensive explanation.
>
> **Response to W2:"The algorithms require careful tuning of hyperparameters, which can be time-consuming."**
>
> Regarding hyperparameter tuning, we agree that the method introduces several new parameters (e.g., momentum decay factor, threshold). That said, we found the method to be **relatively robust** to a wide range of settings, and we provide the default configurations in the supplementary material to facilitate reproducibility. We provide an ablation experiment on hyperparameters in the table below, showing that the convergence of T-BSO and BSO is not sensitive to hyperparameters. The base setting is$(\gamma = 1 \times 10^{-6}, \beta_1 = 1 \times 10^{-3}, \beta_2 = 1 \times 10^{-5})$. As observed from the table, the final convergence result is not significantly affected by variations in hyperparameters, provided they remain within a certain range.
>
> | Model | Base | $\gamma $ |$\beta_1$   | $\beta_2$ |
> |:-----------:|:-------:|:--------------:|:----------------------:|:------------------------:|
> | T-BSO | 75.72 |`75.68(5e-7)`, `74.68(2e-6)` | `74.95(5e-4)`,`75.36(2e-3)` | `75.10(5e-6)`,`75.52(2e-5)` |
>
> **Response to Q1:"The author mentioned that BSO is based on a direct learning method. Why not consider the conversion-based method, but choose the direct training method?"**
>
> We opt for direct training over ANN-to-SNN conversion primarily due to its superior compatibility with online training and streaming data scenarios. While conversion-based methods are effective in static image classification tasks, they generally lack the ability to **learn continuously**, rendering them less suitable for online and event-driven environments. In contrast, direct training enables frame-by-frame updates and localized gradient propagation.
>
> **Response to Q2:"The author points out in the introduction, 'leverage the substantial efficiency advantages in BSNN training'. Specifically, how does the BSO algorithm demonstrate its advantages in BSNN?"**
>
> The BSO algorithm directly associates the update of binary weights with the sign flip induced when the momentum of the gradient exceeds a threshold. This eliminates the need for latent weights, and this flip mechanism is theoretically more energy-efficient.
>
> **Response to Q3:"Does the introduction of adaptive thresholds in the T-BSO version of BSO affect the neural dynamics of the online learning framework?"**
>
> We designed the adaptive threshold mechanism in T-BSO to be biologically plausible and hardware-friendly, updating thresholds based on the momentum of temporal gradient. While it alters the flip condition of weights, it does not fundamentally change the underlying online training framework or disrupt spiking dynamics. Instead, it allows the network to **dynamically adjust** its sensitivity to incoming signals, enhancing robustness and efficiency. We also observed that adaptive thresholds help stabilize training and improve generalization across varying input distributions.

---

> > ### Comment · Reviewer_AqeW · 2025-04-06
> >
> > Thank you for the response. The authors have addressed my concerns well.

---

### Official Review · Reviewer_Q2xs · 2025-03-14

**Overall Recommendation:** 4

**Summary:**

The paper introduces Binary Spiking Online Optimization (BSO), a novel training algorithm for Binary Spiking Neural Networks (BSNNs) that significantly reduces memory overhead during training. The key innovations are two-fold:
(1) making memory requirements independent of time steps, and
(2) eliminating latent weight storage by directly updating binary weights through flip signals triggered when gradient momentum exceeds a threshold.
The authors also propose T-BSO, a temporal-aware variant that captures gradient information across time steps for adaptive threshold adjustment. Through theoretical analysis and experiments on datasets like CIFAR-10, CIFAR-100, ImageNet, and DVS-CIFAR10, the authors show that BSO and T-BSO achieve comparable performance compared to existing methods while substantially reducing training memory costs.

**Claims And Evidence:**

The claims are supported by comparative experiments, theoretical convergence proofs, and ablation studies.

**Essential References Not Discussed:**

No.

**Experimental Designs Or Analyses:**

I carefully examined the experimental design and found it to be generally sound.

**Methods And Evaluation Criteria:**

The proposed methods and evaluation criteria are sound, with the authors using various benchmark datasets (CIFAR-10, CIFAR-100, ImageNet, DVS-CIFAR10) that test their BSO and T-BSO algorithm.

**Other Comments Or Suggestions:**

It is recommended that the author conduct more thorough experiments to verify the effectiveness and efficiency of the method from multiple perspectives.

**Other Strengths And Weaknesses:**

Strengths:
1. The paper presents a novel approach to training BSNNs by integrating online training with a unique gradient momentum-based weight-flipping mechanism, which is a promising solution to existing memory and computational constraints.
2. The authors provide comprehensive theoretical analysis, including detailed convergence proofs for their proposed BSO and T-BSO algorithms.

Weaknesses & Questions:
1.  The T-BSO may introduce some additional computational complexity compared to the base BSO method.
2. While promising, the results are primarily demonstrated on image classification tasks, and the method's performance on other domains remains to be explored.
3. The proposed method introduces additional computational complexity through momentum-based weight flipping and adaptive thresholding.
4. As a efficiency-focused work, why the authors did not compare efficiency metrics such as training or inference time, OP, SOP, or energy consumption.

[1] Bitsnns: Revisiting energy-efficient spiking neural networks
[2] Towards energy efficient spiking neural networks: An unstructured pruning framework
[3] Towards Accurate Binary Spiking Neural Networks: Learning with Adaptive Gradient Modulation Mechanism

**Questions For Authors:**

No.

**Relation To Broader Scientific Literature:**

The paper makes contributions to binary spiking neural network research by proposing an online training algorithm that reduces memory overhead.

**Theoretical Claims:**

I carefully reviewed the convergence proofs in Appendix A (pages 11-12).

---

> ### Author Rebuttal · Authors · 2025-03-31
>
> **Response to W1. "The T-BSO may introduce some additional computational complexity compared to the base BSO method."**
>
> Thank you for your observation. It is true that T-BSO incorporates a lightweight second-order temporal gradient mechanism on top of the basic BSO method, potentially increasing computational overhead. Nonetheless, since this computation is averaged over the time dimension, it substantially enhances the network's **fitting capability** without significantly affecting the training time and memory usage. During inference, the temporal-aware threshold remains constant, thereby incurring only minimal computational overhead.
>
> **Response to W2. "While promising, the results are primarily demonstrated on image classification tasks, and the method's performance on other domains remains to be explored."**
>
> In this study, we primarily focus on image classification tasks to offer a clear and controlled evaluation of the proposed method. However, we acknowledge the importance of exploring the applicability of our approach to other domains, such as **speech recognition**. To this end, we train a ResNet19 model with T-BSO on the Google Speech Command (GSC) dataset [1] for speech recognition. The ResNet19 model with T-BSO was trained for 300 epochs on GSC's 35 categories, achieving an accuracy of 96.12%. This result demonstrates the potential of T-BSO in diverse application scenarios.
>
> **Response to W3. "The proposed method introduces additional computational complexity through momentum-based weight flipping and adaptive thresholding."**
>
> Thank you for pointing this out. In the proposed method, additional components such as momentum-based weight flipping and adaptive thresholds are introduced. Notably, the **latent weight** present in existing BSNN works is eliminated, and the subtraction operation is revised to flip signals, thereby enhancing the method’s lightweight and computationally efficient nature.
>
> The momentum mechanism is based solely on simple element-wise operations, which impose minimal computational overhead, while the adaptive threshold does not necessitate extra backpropagation. Crucially, the increased complexity yields significant advantages in terms of performance enhancement, resulting in a favorable trade-off. Moreover, these additional computational operations are not executed during inference.
>
> **Response to W4. "As a efficiency-focused work, why the authors did not compare efficiency metrics such as training or inference time, OP, SOP, or energy consumption."**
>
> Thank you for your insightful comment. We fully agree that evaluating efficiency using metrics such as training/inference time, OPs, SOPs, or energy consumption would provide a more comprehensive understanding of the proposed method's advantages.
>
> The table below provides a comparative analysis of the proposed method against SOP, OP, and energy with QSNN on the CIFAR-10 and CIFAR-100 datasets. We acknowledge that explicitly considering energy consumption and operation count is crucial for efficiency-oriented research, particularly in the context of spiking neural networks. Experimental results demonstrate that our BSO algorithm outperforms Q-SNN in both SOP and energy, owing to its lower spiking firing rate.
>
> | Dataset    | Algorithm | SOP                                      | OP                                      | Energy                                      |
> |------------|-------|------------------------------------------|-----------------------------------------|---------------------------------------------|
> | CIFAR-10   | BSO   | `0.93M(T=2)`, `1.51M(T=4)`, `1.69M(T=6)` | `1.78M` | `9.01uj(T=2)`, `9.54uj(T=4)`, `9.71uj(T=6)` |
> | CIFAR-10   | QSNN  | `1.70M(T=2)`, `2.80M(T=4)`, `3.88M(T=6)` | `1.78M` | `9.72uj(T=2)`, `10.71uj(T=4)`, `11.69uj(T=6)` |
> | CIFAR-100  | BSO   | `0.96M(T=2)`, `1.78M(T=4)`, `2.19M(T=6)` | `1.87M` | `9.48uj(T=2)`, `10.21uj(T=4)`, `10.59uj(T=6)` |
> | CIFAR-100  | QSNN  | `1.70M(T=2)`, `2.78M(T=4)`, `3.86M(T=6)` | `1.87M` | `10.13uj(T=2)`, `11.10uj(T=4)`, `12.08uj(T=6)` |
>
> [1]Warden, Pete. "Speech commands: A dataset for limited-vocabulary speech recognition." arXiv preprint arXiv:1804.03209 (2018).

---

### Official Review · Reviewer_Tbrf · 2025-03-14

**Overall Recommendation:** 4

**Summary:**

The paper proposes Binary Spiking Online Optimization (BSO) and its temporal variant T-BSO for training Binary Spiking Neural Networks (BSNNs) with reduced memory overhead. The work is well-motivated, technically sound, and demonstrates significant improvements in training efficiency and performance across static and neuromorphic datasets. While the core contributions are compelling, the paper would benefit from expanded discussions on related works and scalability to modern architectures like Transformers. With minor revisions, this work has the potential to advance resource-efficient neuromorphic computing.

**Claims And Evidence:**

Yes, the claims are largely supported by evidence:

Claim 1: BSO/T-BSO reduce memory overhead. Table 1 and Figure 5 convincingly show time-independent memory costs (e.g., 3GB vs. Q-SNN’s linear scaling).

Claim 2: Superior performance. Table 2 validates T-BSO’s accuracy gains (e.g., 94.70% on CIFAR-10 vs. OTTT’s 93.73%), though statistical significance testing (e.g., standard deviations) is missing.

Theoretical claims: Regret bounds (Theorem 4.1) are rigorously proven in Appendix A, assuming convexity and bounded gradients.

**Essential References Not Discussed:**

I suggest add more advanced SNNs models recently.

**Experimental Designs Or Analyses:**

Mostly sound:

Ablations: Table 3 and Figure 4 effectively validate T-BSO’s temporal adaptation.

Statistical rigor: Missing standard deviations in Table 2 and Figure 4 reduce reproducibility.

Baselines: Comparisons with OTTT, NDOT, and Q-SNN are thorough, but exclude recent BSNN works like [Cite concurrent BSNN methods].

**Methods And Evaluation Criteria:**

Yes, the methods and evaluations are appropriate:

Methods: BSO’s flip-signal mechanism (Eq. 12) and T-BSO’s second-order momentum (Eq. 14) are novel and well-justified. The elimination of latent weights (Fig. 2) aligns with BSNN efficiency goals.

Evaluation: Benchmarks (CIFAR, ImageNet, DVS-CIFAR10) are standard. Energy metrics (Table 2) and memory analysis (Fig. 5) effectively highlight efficiency gains.

**Other Comments Or Suggestions:**

None.

**Other Strengths And Weaknesses:**

Strengths:

1. The proposed method is novel.
2. The technique is solid.
3. The experiments are sufficient.

Weakness:

1. There could add more comparison with recent BSNN-online hybrid methods or other related advanced SNNs model.

2. Discuss applicability to spiking Transformers.

**Questions For Authors:**

See weakness.

**Relation To Broader Scientific Literature:**

Strong.

**Theoretical Claims:**

Yes, theoretical analysis is correct under stated assumptions:

---

> ### Author Rebuttal · Authors · 2025-03-31
>
> We sincerely thank the reviewer for their thorough evaluation and valuable feedback. Our responses are provided below.
>
> **Response to Claim 2  "statistical significance testing (e.g., standard deviations) is missing."**
>
> We acknowledge that including standard deviations in Table 2 and Figure 4 would enhance the reproducibility of our results. We will revise the table and figure to include the standard deviations alongside the mean values to provide a clearer representation of the variability and ensure the results are more reproducible.
>
> **Response to Experimental Designs and W1: "cite concurrent BSNN methods and compare with BSNN-online hybrid methods", "There could add more comparison with recent BSNN-online hybrid methods or other related advanced SNNs model."**
>
> Thank you for pointing this out. We appreciate your suggestion to include comparisons with recent BSNN works such as [concurrent BSNN]. We would greatly appreciate it if the reviewer could kindly provide relevant references for these concurrent BSNN methods, which we would be happy to incorporate into our analysis. While our current baseline selection was based on a careful review of related methods at the time of submission, we recognize the importance of including more recent approaches to provide a comprehensive comparison. We consider extending BSO and T-BSO to more complex network structures like transformer-based and recurrent model to adapt to different application scenarios.
>
> **Response to W2 "Discuss applicability to spiking Transformers."**
>
> Regarding the applicability to spiking Transformers, we believe our proposed method can be naturally extended to spiking Transformer-based models. In particular, our proposed T-BSO benefits from the adaptive threshold in the time dimension and can provide better convergence while preserving important **temporal information**. We extend our T-BSO to spikingformer[1] on CIFAR-10 with 4 timestep to prove the applicability. In particular, the binary spikingformer with T-BSO achieved an accuracy of 90.15%, indicating that the T-BSO algorithm can still ensure the **convergence** of binary online training in the transformer-like structure.
>
> [1]Zhou, Chenlin, et al. "Spikingformer: Spike-driven residual learning for transformer-based spiking neural network." arXiv preprint arXiv:2304.11954 (2023).

---

### Decision · Program_Chairs · 2025-05-01

**Decision:**

Accept (poster)

**Comment:**

This paper develops online training of Binary Spiking Neural Networks (BSNNs) and proposes Binary Spiking Online Optimization (BSO), that significantly
reduces training memory while maintaining energy efficiency during the forward pass. Moreover, a temporal-aware variant T-BSO is introduced.
Theoretical analysis shows that the convergence of both BSO and T-BSO algorithms based on the regret bounds. Experiments on datasets like CIFAR-10, CIFAR-100, ImageNet, and DVS-CIFAR10, verify its effectiveness.

The main strengths of this paper are:

- the potential to advance resource-efficient neuromorphic computing
- a novel approach to training BSNNs online with theoretical guarantee
- experiments show the proposed method achieve better accuracy and efficiency than existing methods.

The main weaknesses of this paper are:

- the needs of more comparison with other methods (Reviewer Tbrf, AqeW)
- the needs of more discussion like spiking Transformers and more complex architectures (Reviewer Tbrf, CwUQ), efficiency metrics (Reviewer Q2xs, AqeW, CwUQ), computational complexity (Reviewer Q2xs), convergence behavior under noisy data (Reviewer CwUQ)

During the rebuttal phase, all reviewers acknowledged.  Overall, this paper received a consistent positive recommendation, 4 accept from the reviewers.  All the concerns raised during review were  addressed in the later rebuttal.
After reading all the review, the discussion and the rebuttal, the AC agrees its novelty, its potential and superior performance。 In this case, the AC recommends an Accept.